# Designing transformer oil immersion cooling servers for machine learning and first principle calculations

**Keisuke Takahashi**[1]*, **Itsuki Miyazato**[1], **Satoshi Maeda**[1,2], **Lauren Takahashi**[1]

**1** Department of Chemistry, Hokkaido University, Sapporo, Japan, **2** Institute for Chemical Reaction Design and Discovery (WPI-ICReDD), Hokkaido University, Sapporo, Japan

* keisuke.takahashi@sci.hokudai.ac.jp

## Abstract

A transfomer oil immersion cooling server is designed and constructed for machine learning applications and first principle calculations that are carried out for materials-related research. CPU, motherboard, random access memory, hard disk drive, solid state drive, graphic card, and the power supply unit are submerged into the transformer oil in order to cool the entire system. Benchmark tests reveal that overall performance is improved while performance times for multicore calculations are dramatically improved. Furthermore, calculation times for machine learning with large data sets and density functional theory calculations are shortened during single core calculations. Thus, a transformer oil immersion cooling server is proposed to be an alternative cooling system used for improving the performance of first principle calculations and machine learning.

## Introduction

Computer servers have become increasingly vital within materials science due to the rapid growth and increasing applications of machine learning and first principles calculations towards various research endeavors [1–5]. In particular, first principles calculations require large amounts of parallel computing where multiple central processing units (CPU) are engaged simultaneously while machine learning requires large amounts of input and output from both the CPU and random access memory (RAM) [6–11]. Consequently, these processes generate large amounts of heat within parts such as the CPU, motherboard, random access memory (RAM), hard disk drive (HDD), solid state drive (SDD), graphic card, and even the power supply units. Typically, cooling systems are designed for the CPU as they become the hottest part of the server. Air and liquid cooling are commonly used in cooling systems for removing heat from the CPU. Heat sinks are directly connected to the CPU in order to transfer the heat from the CPU to heat sinks where the heat sinks are then cooled via air flow generated by running fan in the air cooling system. For liquid, a water block is mounted against the CPU where water is circulated by a pump and the water is then cooled by a radiator, thereby transferring the heat generated by the CPU to the water. Although the heat of the CPU is efficiently cooled, other parts like the motherboard, random access memory, and power supply unit are

**Data Availability Statement:** All data are within the paper (listed in Tables I and II and Fig 2).

**Funding:** This study was funded by the Japan Science and Technology Agency (JST) in the form

of a Core Research for Evolutional Science and Technology (CREST) grant to KT (No. JPMJCR17P2) and in the form of a Exploratory Research for Advanced Technology (ERATO) grant to SM (JPMJER1903). The funders had no role in study design, data collection, and analysis, decision to publish, or preparation of the manuscript.

**Competing interests:** The authors have declared that no competing interests exist.

commonly cooled by fans which are still limited to cooling those specific parts. Thus, it would be ideal to implement a cooling systems for the entire server.

Liquid immersion cooling is a progressive way to cool computer servers where the entire motherboard, random access memory, and CPU are commonly submerged into a dielectric liquid. Commonly, fluorocarbon and hydrocarbon based liquids are used. However, durability of the computer parts against the oil as well as the environmental effects of fluorocarbon are of concern, especially as fluorocarbon is considered to have a long atmospheric lifetime. Here, transformer oil is proposed as the base of a liquid immersion cooling system for computer servers as transformer oil has excellent electrical insulating and cooling properties.

## Method

### Benchmark methods

Benchmark tests are performed before and after submerging the computer server into the transformer oil immersion cooling system. FLIR ONE Pro, an infrared camera, is used to measure the temperature of the submerged oil cooling server. Geekbench 5 is used to test both single and multicore performances [12]. In particular, the following performances are tested: AES-XTS, text compression, image compression, navigation, HTML5, SQLite, PDF rendering, text rendering, clang, camera, N-body physics, rigid body physics, Gaussian blur, face detection, horizontal detection, image inpainting, HDS, ray tracing, structure from motion, speech recognition, and machine learning. Scores provided by Geekbench 5 are calculated based on a baseline score of 1000 which is the score of an Intel Core i3-8100. These performances are tested for both single and multicore cases before and after being submerged into the transformer oil.

Benchmark tests for first principles calculations are performed using grid based projector augmented wave (GPAW) method [9]. Relaxation of $H_2$ molecules are calculated using linear combination of atomic orbitals within GPAW method. Exchange correlation of Perdew–Burke–Ernzerhof (PBE) with spin polarization is applied for all calculations [13].

Two types of supervised machine learning are implemented within sciki-learn [10]. Random forest regression (RF) and support vector regression (SVR) are used in order to test machine learning performance before and after submerging the computer server. The number of trees is set to 100 in RF while C and gamma are set to 100 and 0.001 in SVR, respectively. High throughput experimental oxidative coupling method catalysts data consisting of 27,622 data points is used as training data. 41 descriptor variables consist of 37 types of catalysts information created using one hot encoding and 4 types of experimental conditions (temperature, $CH_4$ flow, $O_2$ flow, and Ar flow) are used while $C_2$ yield is set as an objective variable [14]. Cross validation is performed where data set is randomly divided into 80% train data and 20% test data. Calculation time is measured for cross validation of 10 random sets of train and test data.

### Building the computer

A computer server is constructed based on the parts listed in Table 1. AMD Ryzen 5 3400G is chosen as the CPU and consists of 4 cores / 8 threads. The CPU, one terabyte-sized SSD, RAM totaling 32 gigabytes, and 4GB of graphic cards are connected to the motherboard. It is important to mention that none of the cooling devices such as fan and water block are attached to the CPU, leaving CPU in an exposed state. In addition, category 6a lan cable and HDMI cables are connected to the motherboard for wired network and outputing to the display, respectively. The constructed computer server is shown in Fig 1(a). Note that the server is not treated with a protective coating before being submerged in the oil. Before submersion, benchmark

**Table 1. Specification details of parts used for submerging.** CPU: central processing unit, SSD: solid state drive, RAM: random access memory, GPU: graphics processing unit.

|  | Specification |
|---|---|
| CPU | AMD Ryzen 5 3400G |
| Motherboard | ASROCK B450 |
| Power Unit | ANTEC 650W |
| SSD | Crucial SSD 1TB |
| RAM | CORSAIR DDR4-3200 16Gx2 |
| GPU | ASUSGeForce GTX 1650 OC Edition 4GB |

tests are run where benchmark tests using Geekbench 5, first principle calculations, and machine learning are performed without the presence of a cooling system.

The constructed computer server is then submerged into the transformer oil. Transformer oil "high pressure insulating oil A", produced by ENEOS Corporation, is used [15]. Further specifications regarding the transformer oil are collected in Table 2. The chosen transformer oil has low viscosity while possessing high insulating and high antioxidative properties, which can benefit the server. In particular, its low viscosity allows for a high cooling rate while its insulating properties help protect operations by insulating electricity between the server and oil and its antioxidative properties give the oil a longer lifetime. Here, 80 liters of the transformer oil are poured into the computer server as shown in Fig 1(b). Note that all of the server parts– including the motherboard, CPU, RAM, SSD, power supply unit, grpahic card, lan cable, and HDMI cable– are submerged in order to cool the entire server. Both of power supply unit and graphic card have fans. Those fans create the circulation flow within the transformer oil, thus, heat is continuously removed.

Once submerged, the server is operated under the transformer oil. An infrared camera is used to measure the temperature of the transformer oil while under operation as shown in Fig 1(c). Fig 1(c) indicates that the oil temperature is measured to be 28.6˚C. More importantly, not only is the CPU cooled by the oil but all other parts of the server are also simultaneously cooled by the transformer oil, thus, the oil acts as a heat sink. The rise in temperature may affect the viscosity of the oil in a manner that can be seen as beneficial for the server. In particular, the kinematic viscosity of the transformer oil is reported to be 8.09 mm$^2$/s at 40C and decreases to 2.21 mm$^2$/s when at 100C as shown in Table 2. This demonstrates that by increasing temperature, the kinematic viscosity decreases to 1/4 of its original viscosity with an increase of 60C. The server is potentially benefitting this as low viscosity allows for fluid to

**Table 2. The details of the transformer oil [15].**

|  | Specification |
|---|---|
| Density | 0.87kg/l |
| kinematic viscosity(40˚C) | 8.09 mm$^2$/s |
| kinematic viscosity(100˚C) | 2.21 mm$^2$/s |
| Flash point | 150˚C |
| Acid Value | 0.00mgKOH/g |
| Corrosive Sulfur | Non-corrosive |
| Electrical breakdown | 70 kV |
| Dissipation factor | 10% |
| Volume Resistivity(80˚C) | 45TΩ m |
| Benzotriazole | 10/kG |

**(a)**

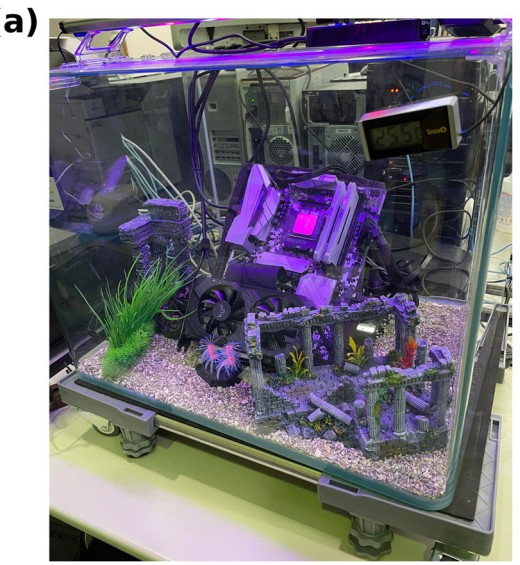

**(b)**

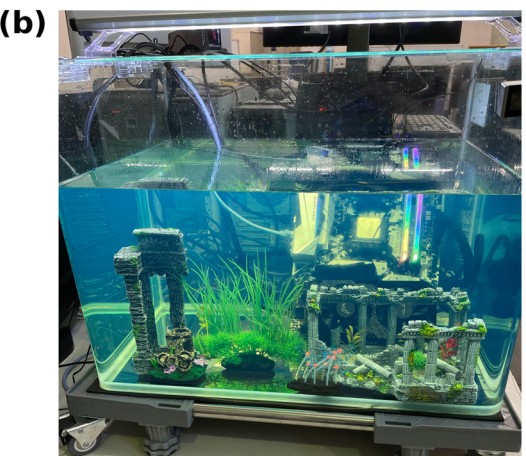

**(c)**

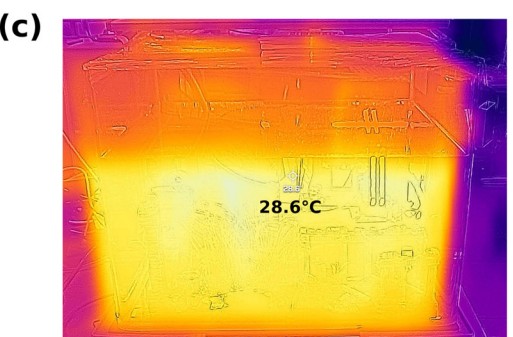

**Fig 1.** The constructed computer server (a) without the presence of the transformer oil and (b) fully submerged in the transformer oil. (c) Thermal imaging of the submerged computer server under operation.

circulate much easier when compared to high viscosity. As the server runs and generates heat, the viscosity of the oil decreases and improves fluid circulation, which can thus help remove the heat from the server and help keep the server cool. These results thereby demonstrate that the constructed computer is operational while immersed within the transformer oil.

## Benchmark

Benchmark tests of the constructed computer server are performed before and after being submerged into the transformer oil. Single core performance is first evaluated across 22 different categories as shown in Fig 2(a) which compared scores from before and after being submerged into transformer oil.

In general, one can see from Fig 2(a) that single core performance slightly increased for most test categories when the server is submerged in the transformer oil. When comparing the benchmark results of the multi-cores, however, performance differences become much more obvious. As seen in Fig 2(b), multi-core performance is dramatically increased once the server is submerged. Considering that multi-core calculations can generate more heat than single cores, one can believe that the transformer oil can contribute towards lowering the sever temperature, thereby resulting in better performance. In the same fashion, density functional theory (DFT) calculations, random forest (RF), and support vector regression (SVR) are evaluated using a single core and shown in Fig 2(c). Relaxation of a $H_2$ molecule is performed where it takes 29.8 seconds to complete the relaxation calculation without the transformer oil while it takes 26.1 seconds to complete the relaxation calculation with the presence of transformer oil. Here, one can see that even with single core calculations, submerging the computer server into the transformer oil has shortened the time taken for density functional theory calculations by 3.7 seconds.

From here, performance of data science techniques are evaluated. 27,622 data points consisting of 41 descriptor variables and 1 objective variable are trained using RF and SVR. For the case of RF, conducting cross-validation with the exposed computer server takes 77.9 seconds to complete while the same cross-validation takes 67.0 second when the server is submerged. Submerging the server into the oil thus reduces completion time by 10.9 seconds. This effect becomes more pronounced in the case of SVR, which requires larger computational times than RF. For the case of SVR, the submitted job is completed within 691.3 seconds with the exposed computer server while the submerged server completes the submitted job within 336.3 seconds. By submerging the server into the transformer oil, it becomes very clear that the time required to complete said task is reduced by 336.0 seconds. This result suggests that within the same timeframe, the submerged server can potentially complete two jobs by the time the exposed server finishes its first job. The drastic decrease in calculation time can be attributed to the transformer oil, which is able to cool the large amount of heat generated by the single-core heavy calculation. With these results, one can conclude that the transformer oil can be an effective cooling system for single-core heavy calculations as well as multi-core calculations that are carried out during first princicples calculations and machine learning calculations.

## Conclusion

A transformer oil immersion cooling server is proposed for first principles calculations and machine learning. A transformer oil with high insulative and high antioxidative properties is chosen where all server parts- including CPU, RAM, motherboard, power unit, graphic card, and all related cabled- are submerged into the transformer oil. The submerged server is successfully operational upon being submerged where the temperature of the oil during server operation remains at 28.6˚C. It must be noted that server lifetime within the transformer oil must still be evaluated as there is possibility of the oil corroding the cables and plastics within the server. One can consider that one of the drawbacks of the transformer oil could be oxidation. As time passes, oxidation can potentially affect the durability of the oil. Further study is required to better understand the effects oxidation can have upon performance and durability

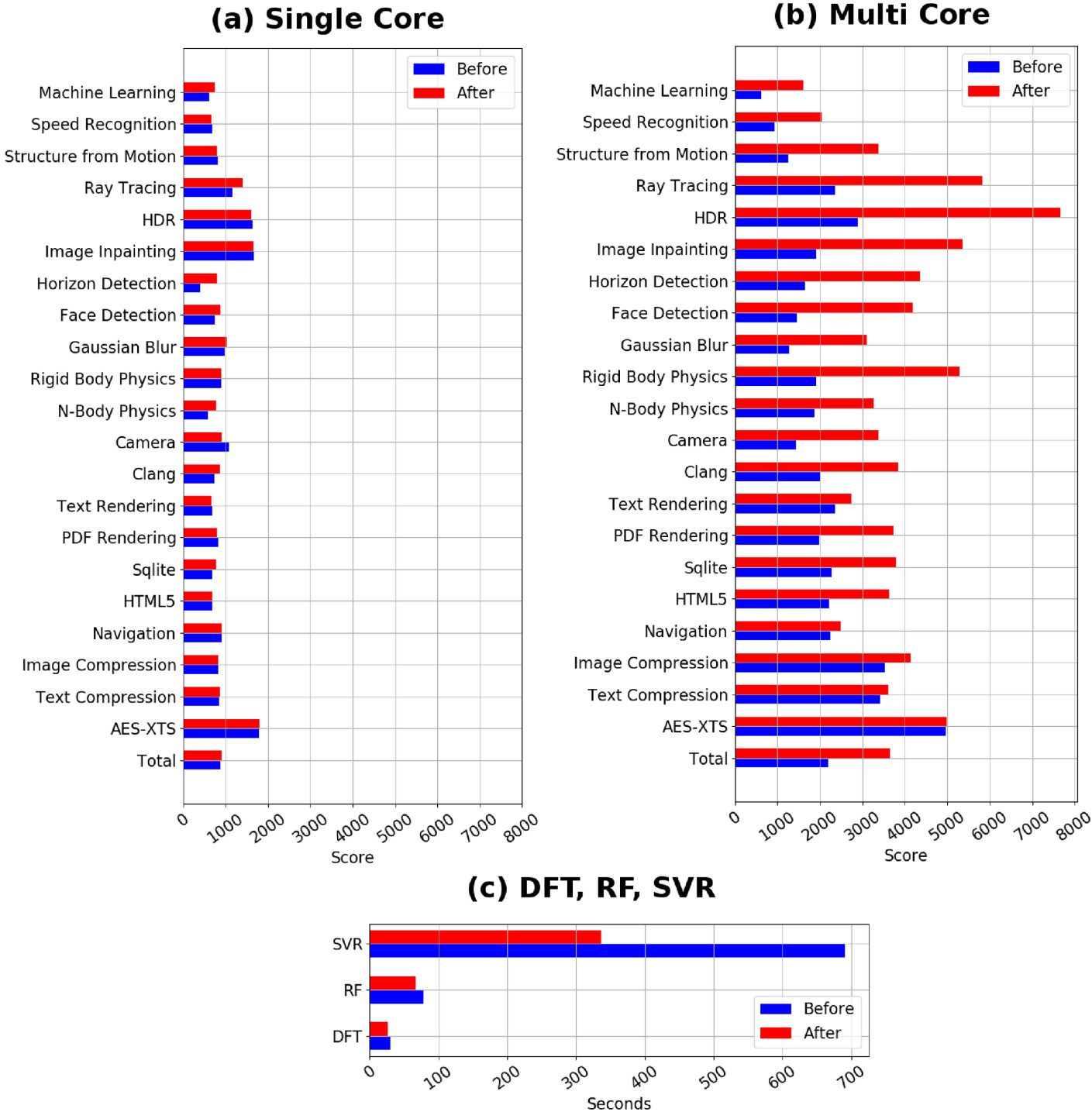

**Fig 2.** Benchmark tests by Geekbench 5 with and without the transformer oil (a) single core and (b) multi core test results by Geekbench 5, and (c) density functional theory (DFT), random forest (RF), and support vector machine (SVR) single core calculations where computational time is evaluated with and without the presence of transformer oil.

of the oil over a longer period of time Benchmark tests demonstrate that single-core performance is moderately improved and multi-core performance is greatly improved when submerged within the transformer oil. Additionally, calculation times for first principles calculations and machine learning applications are shortened by the transformer oil, where this effect is magnified for the case of heavy calculations presented by SVR. Thus, a transformer oil immersion cooling server is designed and demonstrated to be an effective cooling option for servers that conduct first principles calculations and machine learning calculations.

## Author Contributions

**Conceptualization:** Keisuke Takahashi, Lauren Takahashi.

**Data curation:** Keisuke Takahashi.

**Formal analysis:** Keisuke Takahashi.

**Funding acquisition:** Keisuke Takahashi, Satoshi Maeda.

**Investigation:** Keisuke Takahashi, Itsuki Miyazato, Lauren Takahashi.

**Methodology:** Keisuke Takahashi, Lauren Takahashi.

**Project administration:** Keisuke Takahashi.

**Resources:** Keisuke Takahashi, Satoshi Maeda.

**Software:** Keisuke Takahashi, Lauren Takahashi.

**Supervision:** Keisuke Takahashi.

**Validation:** Keisuke Takahashi, Lauren Takahashi.

**Visualization:** Keisuke Takahashi.

**Writing – original draft:** Keisuke Takahashi.

**Writing – review & editing:** Keisuke Takahashi.

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
