## [Decision Letter · Decision Letter 0]

20 Jan 2022

PONE-D-21-35184Designing Transformer Oil Immersion Cooling Servers for Machine Learning and First Principle CalculationsPLOS ONE

Dear Dr. Takahashi,

Thank you for submitting your manuscript to PLOS ONE. After careful consideration, we feel that it has merit but does not fully meet PLOS ONE’s publication criteria as it currently stands. Therefore, we invite you to submit a revised version of the manuscript that addresses the points raised during the review process.

We look forward to receiving your revised manuscript.

Kind regards,

M. Ali Haider, Ph.D.

Academic Editor

PLOS ONE

Journal Requirements:

“This work is funded by Japan Science and Technology Agency(JST) CREST Grant Number JPMJCR17P2. This work is also funded by Japan Science and Technology Agency(JST) ERATO Grant Number JPMJER1903, and JSPS-WPI.”

“This work is funded by Japan Science and Technology Agency(JST) CREST Grant Number JPMJCR17P2. This work is also funded by Japan Science and Technology Agency(JST) ERATO Grant Number JPMJER1903, and JSPS-

WPI.”

 “This work is funded by Japan Science and Technology Agency(JST) CREST Grant Number JPMJCR17P2. This work is also funded by Japan Science and Technology Agency(JST) ERATO Grant Number JPMJER1903, and JSPS-

WPI.”

Reviewers' comments:

Reviewer's Responses to Questions

**Comments to the Author**

1. Is the manuscript technically sound, and do the data support the conclusions?

Reviewer #1: Yes

Reviewer #2: Yes

2. Has the statistical analysis been performed appropriately and rigorously? 

Reviewer #1: Yes

Reviewer #2: Yes

3. Have the authors made all data underlying the findings in their manuscript fully available?

Reviewer #1: No

Reviewer #2: Yes

4. Is the manuscript presented in an intelligible fashion and written in standard English?

Reviewer #1: Yes

Reviewer #2: Yes

5. Review Comments to the Author

Reviewer #1: Computational servers are important in almost all fields of science and technology. These servers generate large amount of heat. Efficient cooling is required to keep the servers functional for a longer period of time. The performance of heated servers usually goes down. In this paper, authors have put their efforts in solving the issue of heated servers and have tried to use transformer oil as a heat sink. Authors have reported that speed of first principle calculations and machine learning applications are better in servers which are immersed in transformer oils. This is really a nice piece of work however authors are requested to clarify few doubts:

• Authors report that the server is functional in the oil. Did they use any type of coating on the server to protect the machine?

• Authors see an improvement in server’s performance after keeping it inside the transformer oil. Authors should explain the reason behind this finding.

• Authors get better results in multi-core machines. The reason behind this observation is missing.

• It is also interesting to know if the same oil can be recycled again and for how many times.

Reviewer #2: Abstract: CPU, motherboard, ...... are submerged into the transformer oil.

Authors intent here is not clear. Need to reconstructed.

What is feasibility of transformer oil immersion cooling servers in real world considering implementation challenges?

Introduction:

unnecessary citations for will known facts should be avoided.

Methods:

It is not clear how heat from transformer oil would be removed? All other systems have circulation in place which continuously removes heat but herein no such details are mentioned.

Experimental results could have been explained more especially effect of heating on density and viscosity. What is the effect of rise in temperature on the physical properties of the fluid.

6. PLOS authors have the option to publish the peer review history of their article (what does this mean?). If published, this will include your full peer review and any attached files.

Reviewer #1: No

Reviewer #2: No

---

## [Author Response · Author response to Decision Letter 0]

9 Mar 2022

Dear Editors of PLOS ONE,

I have carried out revisions throughout the manuscript on the basis of the reviewers' comments. Comments to the reviewers are colored in red and mentioned after =>.

If there are any questions or concerns, please do not hesitate to contact me. I'm more than happy to revise the manuscript.

Best regards,

Keisuke Takahashi

Ph.D. Keisuke Takahashi

Department of Chemistry, 

Hokkaido University, 

North 10, West 8, Sapporo 060-8510, Japan

Here is the answers to reviewers comment.

Editorial Edit:

=> Fixed

2- Please note that funding information should not appear in the Acknowledgments section or other areas of your manuscript. We will only publish funding information present in the Funding Statement section of the online submission form.

=> Fixed

3-Please state what role the funders took in the study.  If the funders had no role, please state: "The funders had no role in study design, data collection and analysis, decision to publish, or preparation of the manuscript."

=> Contribution is now witten

4- Please review your reference list to ensure that it is complete and correct. If you have cited papers that have been retracted, please include the rationale for doing so in the manuscript text, or remove these references and replace them with relevant current references. Any changes to the reference list should be mentioned in the rebuttal letter that accompanies your revised manuscript. If you need to cite a retracted article, indicate the article’s retracted status in the References list and also include a citation and full reference for the retraction notice.

=> Confirmed

Reviewer 1:

Computational servers are important in almost all fields of science and technology. These servers generate large amount of heat. Efficient cooling is required to keep the servers functional for a longer period of time. The performance of heated servers usually goes down. In this paper, authors have put their efforts in solving the issue of heated servers and have tried to use transformer oil as a heat sink. Authors have reported that speed of first principle calculations and machine learning applications are better in servers which are immersed in transformer oils. This is really a nice piece of work however authors are requested to clarify few doubts:

• Authors report that the server is functional in the oil. Did they use any type of coating on the server to protect the machine?

=> We haven't applied any coating. We added the following sentence.

 “Note that the server is not treated with a protective coating before being submerged in the oil.”

• Authors see an improvement in server’s performance after keeping it inside the transformer oil. Authors should explain the reason behind this finding.

=> We believe that the transformer oil act as a heat sink.We added the following sentence.

“More importantly, not only is the CPU cooled by the oil but all other parts of the server are also simultaneously cooled by the transformer oil, thus, the oil acts as a heat sink.”

• Authors get better results in multi-core machines. The reason behind this observation is missing. The multi-core process generates more heat than the single core process. From this, we can consider the effect of cooling to be more crucial. We have added the following to the manuscript. 

" Considering that multi-core calculations can generate more heat than single cores, one can believe that the transformer oil can contribute towards lowering the sever temperature, thereby resulting in better performance.."

• It is also interesting to know if the same oil can be recycled again and for how many times.

=>One of the drawbacks of the transformer oil is oxidation. As time passes, oxidation can potentially affect the durability of the oil. We have added the following sentence to the manuscript. "It must be noted that server lifetime within the transformer oil must still be evaluated as there is possibility of the oil corroding the cables and plastics within the server."

Reviewer 2:

 Abstract: CPU, motherboard, ...... are submerged into the transformer oil.

Authors intent here is not clear. Need to reconstructed.

=> We meant that we wanted to cool the entire system where normally CPU is only cooled. We have added the following sentence to the abstruct.

“CPU, motherboard, random access memory, hard disk drive, solid state drive, graphic card, and the power supply unit are submerged into the transformer oil in order to cool the entire system.”

What is feasibility of transformer oil immersion cooling servers in real world considering implementation challenges?

=> The challenge would be lifetime and oxidation. One of the drawbacks of the transformer oil is oxidation. As time passes, oxidation can potentially affect the durability of the oil. We have added the following sentence to the manuscript. "It must be noted that server lifetime within the transformer oil must still be evaluated as there is possibility of the oil corroding the cables and plastics within the server. One of the drawbacks of the transformer oil could be oxidation. As time passes, oxidation can potentially affect the durability of the oil. Further study is required to better understand the effects oxidation can have upon performance and durability of the oil over a longer period of time"

Introduction:

unnecessary citations for will known facts should be avoided.

=> All citations at introduction are about materials informatics and computational materials science/Chemistry/physics now.

Methods:

It is not clear how heat from transformer oil would be removed? All other systems have circulation in place which continuously removes heat but herein no such details are mentioned.

=> Thank you for pointing this out. Both of power supply unit and graphic card have fans. Those fans create the circulation flow within the transformer oil, thus, heat is continuously removed.We have added the following to the manuscript.

“Both of power supply unit and graphic card have fans. Those fans create the circulation flow within the transformer oil, thus, heat is continuously removed.”

Experimental results could have been explained more especially effect of heating on density and viscosity. What is the effect of rise in temperature on the physical properties of the fluid.

=> It is quite challenging to identify exact physical properies of the fluid. However, as reviewers suggested, kinematic viscosity explains this well. In particular, kinematic viscosity of the transformer oil in this work is 8.09 mm2/s at 40C while its 2.21 mm2/s at 100C. This shows that kinematic viscosity become approximately ¼ with increase of 60C. In another word, low viscosity enable to circulate the fliud easily compared to high viscosity. Thus, low viscosity upon the rise of tempeature is one of the main feature of this work. The following sentences are now added.

“"The rise in temperature is found to affect the viscosity of the oil in a manner that can be seen as beneficial for the server. To start, the kinematic viscosity of the transformer oil is reported to be 8.09 mm2/s at 40C and decreases to 2.21 mm2/s when at 100C. This demonstrates that by increasing temperature, the kinematic viscosity decreases to 1/4 of its original viscosity with an increase of 60C. The server is potentially benefitting this as low viscosity allows for fluid to circulate much easier when compared to high viscosity. As the server runs and generates heat, the viscosity of the oil decreases and improves fluid circulation, which can thus help remove the heat from the server and help keep the server cool. "”

---

## [Editor Report · Decision Letter 1]

30 Mar 2022

Designing Transformer Oil Immersion Cooling Servers for Machine Learning and First Principle Calculations

PONE-D-21-35184R1

Dear Dr. Takahashi,

We’re pleased to inform you that your manuscript has been judged scientifically suitable for publication and will be formally accepted for publication once it meets all outstanding technical requirements.

Kind regards,

Mohammad Ali Haider, Ph.D.

Academic Editor

PLOS ONE
---

## [Editor Report · Acceptance letter]

12 Apr 2022

PONE-D-21-35184R1 

Designing Transformer Oil Immersion Cooling Servers for Machine Learning and First Principle Calculations 

Dear Dr. Takahashi:

I'm pleased to inform you that your manuscript has been deemed suitable for publication in PLOS ONE. Congratulations! Your manuscript is now with our production department. 

Kind regards, 

on behalf of

Dr. Mohammad Ali Haider 

Academic Editor

PLOS ONE